# Maximum Expected Hitting Cost of a Markov Decision Process and Informativeness of Rewards

**Falcon Z. Dai**
Toyota Technological Institute at Chicago
Chicago, IL, USA 60637
`dai@ttic.edu`

**Matthew R. Walter**
Toyota Technological Institute at Chicago
Chicago, IL, USA 60637
`mwalter@ttic.edu`

## Abstract

We propose a new complexity measure for Markov decision processes (MDPs), the *maximum expected hitting cost* (MEHC). This measure tightens the closely related notion of diameter [JOA10] by accounting for the reward structure. We show that this parameter replaces diameter in the upper bound on the optimal value span of an extended MDP, thus refining the associated upper bounds on the regret of several `UCRL2`-like algorithms. Furthermore, we show that potential-based reward shaping [NHR99] can induce equivalent reward functions with varying informativeness, as measured by MEHC. We further establish that shaping can reduce or increase MEHC by at most a factor of two in a large class of MDPs with finite MEHC and unsaturated optimal average rewards.

## 1 Introduction

In the average reward setting of reinforcement learning (RL) [Put94; SB98], an algorithm learns to maximize its average rewards by interacting with an *unknown* Markov decision process (MDP). Similar to analysis in multi-armed bandits and other online machine learning problems, (cumulative) regret provides a natural model to evaluate the efficiency of a learning algorithm. With the `UCRL2` algorithm, Jaksch, Ortner, and Auer [JOA10] show a problem-dependent bound of $\tilde{O}(DS\sqrt{AT})$ on regret and an associated logarithmic bound on the expected regret, where $D$ is the diameter of the actual MDP (Definition 1), $S$ the size of the state space, and $A$ the size of the action space. Many subsequent algorithms [FLP19] enjoy similar diameter-dependent bounds. This establishes diameter as an important measure of complexity for an MDP. However, strikingly, this measure is independent of rewards and is a function of only the transitions. This is obviously peculiar as two MDPs differing only in their rewards would have the same regret bounds even if one gives the maximum reward for all transitions. We review the related key observation by Jaksch, Ortner, and Auer [JOA10], and refine it with a new lemma (Lemma 1), establishing a reward-*sensitive* complexity measure that we refer to as the *maximum expected hitting cost* (MEHC, Definition 2), which tightens the regret bounds of `UCRL2` and similar algorithms by replacing diameter (Theorem 1).

Next, with respect to this new complexity measure, we describe a notion of reward informativeness (Section 2.4). Intuitively speaking, in an environment, the *same* desired policies can be motivated by different (immediate) rewards. These differing definitions of rewards can be more or less *informative* of useful actions, i.e., yielding high long-term rewards. To formalize this intuition, we study a way to reparametrize rewards via potential-based reward shaping (PBRS) [NHR99] that can produce different rewards with the same near-optimal policies (Section 2.5). We show that the MEHC changes under reparametrization by PBRS and, in turn, so do regret and sample complexity, substantiating this notion of informativeness. Lastly, we study the extent of its impact. In particular, we show that there is a factor-of-two limit on its impact on MEHC in a large class of MDPs (Theorem 2). This result and the concept of reward informativeness may be useful for a task designer crafting a reward function (Section 3). The detailed proofs are deffered to Appendix A.

The main contributions of this work are two-fold:

- We propose a new MDP structural parameter, maximum expected hitting cost (MEHC), that accounts for both transitions and rewards. This parameter replaces diameter in the regret bounds of several model-based RL algorithms.

- We show that potential-based reward shaping can change the maximum expected hitting cost of an MDP and thus the regret bound. This results in a set of equivalent MDPs with different learning difficulties as measured by regret. Moreover, we show that their MEHCs differ by a factor of at most two in a large class of MDPs.

## 1.1 Related work

This work is closely related to the study of diameter as an MDP complexity measure [JOA10], which is prevalent in the regret bounds of RL algorithms in the average reward setting [FLP19]. As noted by Jaksch, Ortner, and Auer [JOA10], unlike some previous measures of MDP complexity such as the return mixing time [KS02; BT02], diameter depends only on the transitions, but not the rewards. The core reason for the presence of diameter in the regret analysis is that it upper bounds the optimal value span of the extended MDP that summarizes the observations (Section 2.3 and Equation 8). We review and update this observation with a reward-dependent parameter we called maximum expected hitting cost (Lemma 1). Interestingly, the gap between diameter and MEHC can be arbitrarily large $\kappa(M) \leq r_{\max} D(M)$; there are MDPs with finite MEHC and infinite diameter. These MDPs are non-communicating but have saturated optimal average rewards $\rho^*(M) = r_{\max}$. Intuitively, there is a state $s$ in these MDPs from which the learner cannot visit some other state $s'$, but can nonetheless achieve the maximum possible average reward, thus allowing for good regret guarantees; the unreachable states will not seem better than the reachable ones under the principle of optimism in the face of uncertainty (OFU). We will use UCRL2 [JOA10] as an example algorithm throughout the rest of the article, however the main results do not depend on it. In particular, with MEHC, its regret bounds are updated (Theorem 1).

Another important comparison is with optimal bias span [Put94; BT09; Fru+18], a reward-dependent parameter of MDPs. Here, we again find that the gap can be arbitrarily large $sp(M) \leq \kappa(M)$.[1] These non-communicating MDPs would have unsaturated optimal average reward $\rho^*(M) < r_{\max}$. But as shown elsewhere [FPL18; Fru+18], extra knowledge of (some upper bound on) the optimal bias span is necessary for an algorithm to enjoy a regret that scales with this smaller parameter. In contrast, UCRL2, which scales with MEHC, does not need to know the diameter or MEHC of the actual MDP.

Potential-based reward shaping [NHR99] was originally proposed as a *solution technique* for a programmer to influence the sample complexity of their reinforcement learning algorithm, without changing the near-optimal policies in episodic and discounted settings. Prior theoretical analysis involving PBRS [NHR99; Wie03; WCE03; ALZ08; Grz17] mostly focuses on the consistency of RL against the shaped rewards, i.e., the resulting learned behavior is also (near-)optimal in the original MDP, while suggesting empirically that the sample complexity can be changed by a well specified potential. In this work, we use PBRS to construct $\Pi$-equivalent reward functions in the average reward setting (Section 2.4) and show that two reward functions related by a shaping potential can have different MEHCs, and thus different regrets and sample complexities (Section 2.5). However, a subtle but important technical requirement of $[0, r_{\max}]$-boundedness of MDPs makes it difficult to immediately apply our results (Section 2.5 and Theorem 2) to the treatment of PBRS as a solution technique because an arbitrary potential function picked without knowledge of the original MDP may not preserve the $[0, r_{\max}]$-boundedness. Nevertheless, we think our work may bring some new perspectives to this topic.

## 2 Results

### 2.1 Markov decision process

A *Markov decision process* is defined by the tuple $M = (\mathcal{S}, \mathcal{A}, p, r)$, where $\mathcal{S}$ is the state space, $\mathcal{A}$ is the action space, $p : \mathcal{S} \times \mathcal{A} \to \mathcal{P}(\mathcal{S})$ is the transition function, and $r : \mathcal{S} \times \mathcal{A} \to \mathcal{P}([0, r_{\max}])$ is the reward function with mean $\bar{r}(s, a) := \mathbb{E}[r(s, a)]$. We assume that the state and action spaces are finite, with sizes $S := |\mathcal{S}|$ and $A := |\mathcal{A}|$, respectively. At each time step $t = 0, 1, 2, \ldots$, an algorithm $\mathfrak{L}$ chooses an action $a_t \in \mathcal{A}$ based on the observations up to that point. The state transitions to $s_{t+1}$ with probability $p(s_{t+1}|s_t, a_t)$ and a reward $r_t \in [0, r_{\max}]$ is drawn according to the distribution $r(s_t, a_t)$.[2] The transition probabilities and reward function of the MDP are unknown to the learner. The sequence of random variables $(s_t, a_t, r_t)_{t \geq 0}$ forms a stochastic process. Note that a stationary deterministic policy $\pi : \mathcal{S} \to \mathcal{A}$ is a restrictive type of algorithm whose action $a_t$ depends only on $s_t$. We refer to stationary deterministic policies as policies in the rest of the paper.

Recall that in a Markov chain, the *hitting time* of state $s'$ starting at state $s$ is a random variable $h_{s \to s'} := \inf\{t \in \mathbb{N}_{\geq 0} | s_t = s' \text{ and } s_0 = s\}$.[3] [LPW08].

**Definition 1** (Diameter, [JOA10]). *Suppose in the stochastic process induced by following a policy $\pi$ in MDP $M$, the time to hit state $s'$ starting at state $s$ is $h_{s \to s'}(M, \pi)$. We define the* diameter *of $M$ to be*

$$D(M) := \max_{s, s' \in \mathcal{S}} \min_{\pi:\mathcal{S} \to \mathcal{A}} \mathbb{E}\left[h_{s \to s'}(M, \pi)\right].$$

We incorporate rewards into diameter, and introduce a novel MDP parameter.

**Definition 2** (Maximum expected hitting cost). *We define the* maximum expected hitting cost *of a Markov decision process $M$ to be*

$$\kappa(M) := \max_{s, s' \in \mathcal{S}} \min_{\pi:\mathcal{S} \to \mathcal{A}} \mathbb{E}\left[\sum_{t=0}^{h_{s \to s'}(M, \pi) - 1} r_{max} - r_t\right].$$

Observe that MEHC is a smaller parameter, that is, $\kappa(M) \leq r_{\max} D(M)$, since for any $s, s', \pi$, we have $r_{\max} - r_t \leq r_{\max}$.

### 2.2 Average reward criterion, and regret

The *accumulated reward* of an algorithm $\mathfrak{L}$ after $T$ time steps in MDP $M$ starting at state $s$ is a random variable

$$R(M, \mathfrak{L}, s, T) := \sum_{t=0}^{T-1} r_t.$$

We define the *average reward* or *gain* [Put94] as

$$\rho(M, \mathfrak{L}, s) := \lim_{T \to \infty} \frac{1}{T} \mathbb{E}\left[R(M, \mathfrak{L}, s, T)\right]. \tag{1}$$

We will evaluate policies by their average reward. This can be maximized by a stationary deterministic policy and we define the *optimal average reward* of $M$ starting at state $s$ as

$$\rho^*(M, s) := \max_{\pi:\mathcal{S} \to \mathcal{A}} \rho(M, \pi, s). \tag{2}$$

Furthermore, we assume that the optimal average reward starting at any state to be the same, i.e., $\rho^*(M, s) = \max_{s'} \rho^*(M, s')$ for any state $s$. This is a natural requirement of an MDP in the online setting to allow for any hope for a vanishing regret. Otherwise, the learner may take actions leading to states with a lower average optimal reward due to ignorance and incur linear regret

when compared with the optimal policy starting at the initial state. In particular, this condition is true for communicating MDPs [Put94] by virtue of their transitions, but this is also possible for non-communicating MDPs with appropriate rewards. We will write $\rho^*(M) \coloneqq \max_{s'} \rho^*(M, s')$.

We will compete with the expected cumulative reward of an optimal policy *on its trajectory*, and define the *regret* of a learning algorithm $\mathfrak{L}$ starting at state $s$ after $T$ time steps as

$$\Delta(M, \mathfrak{L}, s, T) \coloneqq T\rho^*(M) - R(M, \mathfrak{L}, s, T). \tag{3}$$

## 2.3 Optimism in the face of uncertainty, extended MDP, and `UCRL2`

The principle of optimism in the face of uncertainty (OFU) [SB98] states that for uncertain state-action pairs, i.e., those that we have not visited enough up to this point, we should be optimistic about their outcome. The intuition for doing so is that taking reward-maximizing actions with respect to this optimistic model (in terms of both transitions and immediate rewards for these uncertain state-action pairs), we will have no regret if the optimism is well placed and will otherwise quickly learn more about these suboptimal state-action pairs to avoid them in the future. This fruitful idea has been the basis for many model-based RL algorithms [FLP19] and in particular, UCRL2 [JOA10], which keeps track of the statistical uncertainty via upper confidence bounds.

Suppose we have visited a particular state-action pair $(s, a)$ $N(s, a)$-many times. With confidence at least $1 - \delta$, we can establish that a confidence interval for both its mean reward $\bar{r}(s, a)$ and its transition $p(\cdot|s, a)$ from the Chernoff-Hoeffding inequality (or Bernstein, [FPL18]). Let $b(\delta, n) \in \mathbb{R}$ be the $\delta$-confidence bound after observing $n$ i.i.d. samples of a $[0, 1]$-bounded random variable, $\hat{r}(s, a)$ the empirical mean of $r(s, a)$, $\hat{p}(\cdot|s, a)$ the empirical transition of $p(\cdot|s, a)$. The statistically plausible mean rewards are

$$B_\delta(s, a) \coloneqq \left\{ r' \in \mathbb{R} : |r' - \hat{r}(s, a)| \leq r_{\max} b(\delta, N(s, a)) \right\} \cap [0, r_{\max}]$$

and the statistically plausible transitions are

$$C_\delta(s, a) \coloneqq \left\{ p' \in \mathcal{P}(\mathcal{S}) : ||p'(\cdot) - \hat{p}(\cdot|s, a)||_1 \leq b(\delta, N(s, a)) \right\}.$$

We define an *extended MDP* $M^+ \coloneqq (\mathcal{S}, \mathcal{A}^+, p^+, r^+)$ to summarize these statistics [GLD00; SL05; TB07; JOA10], where $\mathcal{S}$ is the same state space as in $M$, the action space $\mathcal{A}^+$ is a union over state-specific actions

$$\mathcal{A}_s^+ \coloneqq \left\{ (a, p', r') : a \in \mathcal{A}, p' \in C_\delta(s, a), r' \in B_\delta(s, a) \right\}, \tag{4}$$

where $\mathcal{A}$ is the same action space in $M$, $p^+$ the transitions according to the selected distribution $p'$

$$p^+\big(\cdot|s, (a, p', r')\big) \coloneqq p'(\cdot), \tag{5}$$

and $r^+$ is the rewards according to the selected mean reward $r'$

$$r^+\big(s, (a, p', r')\big) \coloneqq r'. \tag{6}$$

It is not hard to see that $M^+$ is indeed an MDP with an infinite but compact action space.

By OFU, we want to find an optimal policy for an optimistic MDP within the set of statistically plausible MDPs. As observed in [JOA10], this is equivalent to finding an optimal policy $\pi^+ : \mathcal{S} \to \mathcal{A}^+$ in the extended MDP $M^+$, which specifies a policy in $M$ via $\pi(s) \coloneqq \sigma_1(\pi^+(s))$, where $\sigma_i$ is the projection map onto the $i$-th coordinate (and an optimistic MDP $\widetilde{M} = (\mathcal{S}, \mathcal{A}, \widetilde{p}, \widetilde{r})$ via transitions $\widetilde{p}(\cdot|s, \pi(s)) \coloneqq \sigma_2(\pi^+(s))$ and mean rewards $\widetilde{r}(s, \pi(s)) \coloneqq \sigma_3(\pi^+(s))$ over actions selected by $\pi$[4]).

By construction of the extended MDP $M^+$, $M$ is in $M^+$ with high confidence, i.e., $\bar{r}(s, a) \in B_\delta(s, a)$ and $p(\cdot|s, a) \in C_\delta(s, a)$ for all $s \in \mathcal{S}, a \in \mathcal{A}$. At the heart of UCRL2-type regret analysis, there is a key observation [JOA10, equation (11)] that we can bound the span of optimal values in the *extended* MDP $M^+$ by the diameter of the actual MDP $M$ under the condition that $M$ is in $M^+$. This observation is needed to characterize how good following the "optimistic" policy $\sigma_1(\pi^+)$ in the actual MDP $M$ is. For $i \geq 0$, the *i-step optimal values* $u_i(s)$ of $M^+$ is the expected total reward by

following an optimal non-stationary $i$-step policy starting at state $s \in \mathcal{S}$. We can also define them recursively (via dynamic programming[5])

$$u_0(s) := 0$$

$$u_{i+1}(s) := \max_{(a,p',r') \in \mathcal{A}_s^+} \left[ r^+\big(s, (a, p', r')\big) + \sum_{s'} p^+\big(s'|s, (a, p', r')\big) u_i(s') \right]$$

By (5) and (6)

$$= \max_{(a,p',r') \in \mathcal{A}_s^+} \left[ r' + \sum_{s'} p'(s')\, u_i(s') \right]$$

By (4)

$$= \max_{a \in \mathcal{A}} \left[ \max_{r' \in B_\delta(s,a)} r' + \max_{p' \in C_\delta(s,a)} \sum_{s'} p'(s')\, u_i(s') \right] \tag{7}$$

We are now ready to restate the observation. If $M$ is in $M^+$, which happens with high probability, Jaksch, Ortner, and Auer [JOA10] observe that

$$\max_s u_i(s) - \min_{s'} u_i(s') \le r_{\max} D(M). \tag{8}$$

However, this bound is too conservative because it fails to account for the rewards collected. By patching this, we tighten the upper bound with MEHC.

**Lemma 1** (MEHC upper bounds the span of values). *Assuming that the actual MDP $M$ is in the extended MDP $M^+$, i.e., $\bar{r}(s,a) \in B_\delta(s,a)$ and $p(\cdot|s,a) \in C_\delta(s,a)$ for all $s \in \mathcal{S}, a \in \mathcal{A}$, we have*

$$\max_s u_i(s) - \min_{s'} u_i(s') \le \kappa(M)$$

*where $u_i(s)$ is the $i$-step optimal undiscounted value of state $s$.*

This refined upper bound immediately plugs into the main theorems of [JOA10, equations 19 and 22, theorem 2].

**Theorem 1** (Reward-sensitive regret bound of `UCRL2`). *With probability of at least $1 - \delta$, for any initial state $s$ and any $T > 1$, and $\kappa := \kappa(M)$, the regret of `UCRL2` is bounded by*

$$\Delta(M, \texttt{UCRL2}, s, T)$$
$$\le \sqrt{\frac{5}{8} T \log\left(\frac{8T}{\delta}\right)} + \sqrt{T} + \kappa \sqrt{\frac{5}{2} T \log\left(\frac{8T}{\delta}\right)} + \kappa S A \log_2\left(\frac{8T}{SA}\right)$$
$$+ \left( \kappa \sqrt{14 S \log\left(\frac{2AT}{\delta}\right)} + \sqrt{14 \log\left(\frac{2SAT}{\delta}\right)} + 2 \right)(\sqrt{2} + 1)\sqrt{SAT}$$
$$\le 34 \max\{1, \kappa\} S \sqrt{AT \log\left(\frac{T}{\delta}\right)}.$$

As a corollary, Theorem 1 implies that `UCRL2` offers $O\left(\frac{\kappa^2 S^2 A}{\varepsilon^2} \log \frac{\kappa SA}{\delta \varepsilon}\right)$ sample complexity [Kak03], by inverting the regret bound by demanding that the per-step regret is at most $\varepsilon$ with probability of at least $1 - \delta$ [JOA10, corollary 3]. Similarly, we have an updated logarithmic bound on the expected regret [JOA10, theorem 4], $\mathbb{E}[\Delta(M, \texttt{UCRL2}, s, T)] = O(\frac{\kappa^2 S^2 A \log T}{g})$ where $g$ is the gap in average reward between the best policy and the second best policy.

## 2.4 Informativeness of rewards

Informally, it is not hard to appreciate the challenge imposed by delayed feedback inherent in MDPs, as actions with high immediate rewards do not necessarily lead to a high *optimal* value. Are there different but "equivalent" reward functions that differ in their *informativeness* with the more informative ones being easier to reinforcement learn? Suppose we have two MDPs differing only in their rewards, $M_1 = (\mathcal{S}, \mathcal{A}, p, r_1)$ and $M_2 = (\mathcal{S}, \mathcal{A}, p, r_2)$, then they will have the same diameters $D(M_1) = D(M_2)$ and thus the same diameter-dependent regret bounds from previous works. With MEHC, however, we may get a more meaningful answer.

Firstly, let us make precise a notion of equivalence. We say that $r_1$ and $r_2$ are $\Pi$-*equivalent* if for any policy $\pi : \mathcal{S} \to \mathcal{A}$, its average rewards are the same under the two reward functions $\rho(M_1, \pi, s) = \rho(M_2, \pi, s)$. Formally, we will study the MEHC of a class of $\Pi$-equivalent reward functions related via a potential.

## 2.5 Potential-based reward shaping

Originally introduced by Ng, Harada, and Russell [NHR99], potential-based reward shaping (PBRS) takes a potential $\varphi : \mathcal{S} \to \mathbb{R}$ and defines shaped rewards

$$r_t^\varphi := r_t - \varphi(s_t) + \varphi(s_{t+1}). \tag{9}$$

We can think of the stochastic process $(s_t, a_t, r_t^\varphi)_{t\geq 0}$ being generated from an MDP $M^\varphi = (\mathcal{S}, \mathcal{A}, p, r^\varphi)$ with reward function $r^\varphi : \mathcal{S} \times \mathcal{A} \to \mathcal{P}([0, r_{\max}])$[6] whose mean rewards are

$$\bar{r^\varphi}(s, a) = \bar{r}(s, a) - \varphi(s) + \mathbb{E}_{s' \sim p(\cdot|s,a)}[\varphi(s')].$$

It is easy to check that $r^\varphi$ and $r$ are indeed $\Pi$-equivalent. For any policy $\pi$,

$$\rho(M^\varphi, \pi, s) = \lim_{T\to\infty} \frac{1}{T}\mathbb{E}\left[R(M^\varphi, \pi, s, T)\right]$$

$$= \lim_{T\to\infty} \frac{1}{T}\mathbb{E}\left[\sum_{t=0}^{T-1} r_t^\varphi\right]$$

$$= \lim_{T\to\infty} \frac{1}{T}\mathbb{E}\left[\sum_{t=0}^{T-1} r_t - \varphi(s_t) + \varphi(s_{t+1})\right]$$

By telescoping sums of potential terms over consecutive $t$

$$= \lim_{T\to\infty} \frac{1}{T}\mathbb{E}\left[-\varphi(s_0) + \varphi(s_T) + \sum_{t=0}^{T-1} r_t\right]$$

$$= \lim_{T\to\infty} \frac{1}{T}\left(-\varphi(s) + \mathbb{E}[\varphi(s_T)] + \mathbb{E}[R(M, \pi, s, T)]\right)$$

The first two terms vanish in the limit

$$= \lim_{T\to\infty} \frac{1}{T}\mathbb{E}[R(M, \pi, s, T)]$$

$$= \rho(M, \pi, s). \tag{10}$$

To get some intuition, it is instructive to consider a toy example (Figure 1). Suppose $0 < \beta < \alpha$ and $\epsilon \in (0, 1)$, then the optimal average reward in this MDP is $1 - \beta$, and the optimal stationary deterministic policy is $\pi^*(s_1) := a_2$ and $\pi^*(s_2) := a_1$, as staying in state $s_2$ yields the highest average reward. As the expected number of steps needed to transition from state $s_1$ to $s_2$ and vice versa are both $1/\epsilon$ via action $a_2$, we conclude that $\kappa(M) = \max\{\alpha, \alpha/\epsilon, \beta/\epsilon, \beta\} = \alpha/\epsilon$. Furthermore, notice that taking action $a_2$ in either state transitions to the other state with probability of $\epsilon$, however the immediate rewards are the same as taking the alternative action $a_1$ to stay in the current state—the immediate rewards are not *informative*. We can differentiate the actions better by shaping with a potential of $\varphi(s_1) := 0$ and $\varphi(s_2) := (\alpha-\beta)/2\epsilon$. The shaped mean rewards become, at $s_1$,

$$\bar{r^\varphi}(s_1, a_2) = 1 - \alpha - \varphi(s_1) + \epsilon\varphi(s_2) + (1-\epsilon)\varphi(s_1) = 1 - (\alpha+\beta)/2 > 1 - \alpha = \bar{r^\varphi}(s_1, a_1)$$

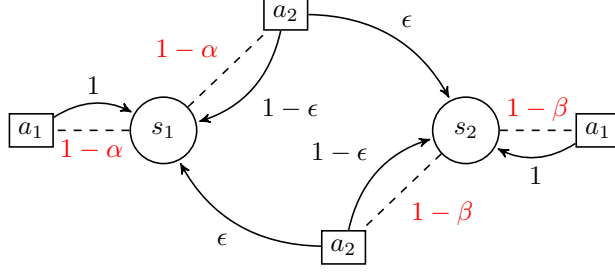

Figure 1: Circular nodes represent states and square nodes represent actions. The solid edges are labeled by the transition probabilities and the dashed edges are labeled by the mean rewards. Furthermore, $r_{\max} = 1$. For concreteness, one can consider setting $\alpha = 0.11, \beta = 0.1, \epsilon = 0.05$.

and at $s_2$,

$$\bar{r}^\varphi(s_2, a_2) = 1 - \beta - \varphi(s_2) + \epsilon\varphi(s_1) + (1 - \epsilon)\varphi(s_2) = 1 - (\alpha+\beta)/2 < 1 - \beta = \bar{r}^\varphi(s_2, a_1).$$

This encourages taking actions $a_2$ at state $s_1$ and discourages taking actions $a_1$ at state $s_2$ simultaneously. The maximum expected hitting cost becomes smaller

$$\kappa(M^\varphi) = \max\left\{\alpha, \beta, \varphi(s_1) - \varphi(s_2) + \frac{\alpha}{\epsilon}, \ \varphi(s_2) - \varphi(s_1) + \frac{\beta}{\epsilon}\right\}$$

$$= \max\left\{\alpha, \beta, \frac{\alpha + \beta}{2\epsilon}, \frac{\alpha + \beta}{2\epsilon}\right\}$$

$$= \frac{\alpha + \beta}{2\epsilon}$$

$$< \frac{\alpha}{\epsilon} = \kappa(M).$$

In this example, MEHC is halved at best when $\beta$ is made arbitrarily close to zero. Noting that the original MDP $M$ is equivalent to $M^\varphi$ shaped with potential $-\varphi$, i.e. $M = (M^\varphi)^{-\varphi}$ from (9), we see that MEHC can be almost doubled. It turns out that halving or doubling the MEHC is the most PBRS can do in a large class of MDPs.

**Theorem 2** (MEHC under PBRS). *Given an MDP $M$ with finite maximum expected hitting cost $\kappa(M) < \infty$ and an unsaturated optimal average reward $\rho^*(M) < r_{max}$, the maximum expected hitting cost of any PBRS-parameterized MDP $M^\varphi$ is bounded by a multiplicative factor of two*

$$\frac{1}{2}\kappa(M) \le \kappa(M^\varphi) \le 2\kappa(M).$$

The key observation is that the expected total rewards along a loop remains unchanged by shaping, which originally motivated PBRS [NHR99]. To see this, consider a loop as a concatenation of two paths, one from $s$ to $s'$ and the other from $s'$ to $s$. Under the shaping of a potential $\varphi$, the expected total rewards of the former is increased by $\varphi(s') - \varphi(s)$ and the latter is decreased by the same amount. For more details, see Appendix A.2.

## 3  Discussion

If we view RL as an engineering tool that "compiles" an arbitrary reward function into a behavior (as represented by a policy) in an environment, then a programmer's primary responsibility would be to craft a reward function that faithfully expresses the intended goal. However, this problem of reward design is complicated by practical concerns for the difficulty of learning. As recognized by Kober, Bagnell, and Peters [KBP13, section 3.4],

> "[t]here is also a trade-off between the complexity of the reward function and the complexity of the learning problem."

Accurate rewards are often easy to specify in a sparse manner (reaching a position, capturing the king, etc), thus hard to learn, whereas dense rewards, providing more feedback, are harder to specify accurately, leading to incorrect trained behaviors. The recent rise of deep RL also exposes "bugs" in some of these designed rewards [CA16]. Our results show that the informativeness of rewards, an aspect of "the complexity of the learning problem" can be controlled to some extent by a well specified potential without inadvertently changing the intended behaviors of the original reward. Therefore, we propose to separate the definitional concern from the training concern. Rewards should be first defined to faithfully express the intended task, and then any extra knowledge can be incorporated via a shaping potential to reduce the sample complexity of training to obtain the same desired behaviors. That is not to say that it is generally easy to find a helpful potential making the rewards more informative.

Though Theorem 2 might be a disappointing result for PBRS, we wish to emphasize that this result most directly concerns algorithms whose regrets scale with MEHC, such as UCRL2. It is conceivable that in a different setting such as discounted total rewards, or for a different RL algorithm, such as SARSA with epsilon-greedy exploration [NHR99, footnote 4], PBRS might have a greater impact on the learning efficiency.

### Acknowledgments

This work was supported in part by the National Science Foundation under Grant No. 1830660. We thank Avrim Blum for many insightful comments. In particular, his challenge to finding a better example has led to Theorem 2. We also thank Ronan Fruit for a discussion on a concept similar to the proposed maximum expected hitting cost that he independently developed in his thesis draft.

## Footnotes

[1]This inequality can be derived as a consequence of Lemma 1 as $N(s, a) \to \infty$, $M^+$ has very tight confidence intervals around the actual transition and mean rewards of $M$. Observe that the span of $u_i$ is equal to $sp(M)$ at the limit of $i \to \infty$ [JOA10, remark 8].

[2]It is important to assume that the support of rewards lies in a *known* bounded interval, often $[0, 1]$ by convention. This is sometimes referred to as a *bounded* MDP in the literature. Analogous to bandits, the details of the reward distribution often is unimportant, and it suffices to specify an MDP with the mean rewards $\bar{r}$.

[3]0-indexing ensures that $h_{s \to s} = 0$. Note also that by convention, $\inf \varnothing = \infty$.

[4] We can set transitions and mean rewards over actions $a \neq \pi(s)$ to $\hat{p}$ and $\hat{r}$, respectively.

[5]In fact, the exact maximization of Equation 7 can be found via extended value iteration [JOA10, section 3.1]

[6]One needs to ensure that $\varphi$ respects the $[0, r_{\max}]$-boundedness of $M$.

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
