[Supplementary Material · hitting-cost-appendix.pdf]

# A   Detailed proofs

## A.1   Proof of Lemma 1

Assuming that the actual MDP $M$ is in the extended MDP $M^+$, i.e., $\bar{r}(s,a) \in B_\delta(s,a)$ and $p(\cdot|s,a) \in C_\delta(s,a)$ for all $s \in \mathcal{S}, a \in \mathcal{A}$, we have

$$\max_s u_i(s) - \min_{s'} u_i(s') \leq \kappa(M)$$

where $u_i(s)$ is the $i$-step optimal undiscounted value of state $s$.

*Proof.* By assumption, the actual mean rewards $\bar{r}$ and transitions $p$ are contained in the extended MDP $M^+$, i.e., for any $s \in \mathcal{S}$ and $a \in \mathcal{A}$, $\bar{r}(s,a) \in B_\delta(s,a)$ and $p(\cdot|s,a) \in C_\delta(s,a)$. Thus for any policy $\pi : \mathcal{S} \to \mathcal{A}$ in the actual MDP $M$, we can construct a corresponding policy $\pi^+ : \mathcal{S} \to \mathcal{A}^+$ in the extended MDP $M^+$

$$\pi^+(s) := \Big(\pi(s), p(\cdot|s,\pi(s)), \bar{r}(s,\pi(s))\Big).$$

Following $\pi^+$ in $M^+$ induces the same stochastic process $(s_t, a_t, r_t)_{t \geq 0}$ as following $\pi$ in $M$. In particular they have the same expected hitting times and expected rewards. By definition $u_i(s)$ is the value of following an *optimal* $i$-step non-stationary policy starting at $s$ in the extended MDP $\mathcal{M}^+$. For any $s'$, by optimality, $u_i(s)$ must be no worse than first following $\pi^+$ from $s$ to $s'$ and then following the optimal $i$-step non-stationary policy from $s'$ onward. Along the path from $s$ to $s'$, we receive rewards according to $\sigma_3(\pi^+) = \bar{r}$ and after arriving at $s'$, we have missed at most $r_{\max} h_{s \to s'}(M^+, \pi^+)$-many rewards of $u_i(s')$ so in expectation

$$u_i(s) \geq \mathbb{E}\left[\sum_{t=0}^{h_{s \to s'}(M^+,\pi^+)-1} r_t\right] + u_i(s') - \mathbb{E}[r_{\max} h_{s \to s'}(M^+, \pi^+)]$$

$$= \mathbb{E}\left[\sum_{t=0}^{h_{s \to s'}(M^+,\pi^+)-1} r_t - r_{\max}\right] + u_i(s')$$

By definition of $\pi^+$, hitting time $h_{s \to s'}(M, \pi) = h_{s \to s'}(M^+, \pi^+)$

$$= \mathbb{E}\left[\sum_{t=0}^{h_{s \to s'}(M,\pi)-1} r_t - r_{\max}\right] + u_i(s').$$

Moving the terms around and we get

$$u_i(s') - u_i(s) \leq \mathbb{E}\left[\sum_{t=0}^{h_{s \to s'}(M,\pi)-1} r_{\max} - r_t\right].$$

Since this holds for any $\pi$ by optimality, we can choose one with the smallest expected hitting cost

$$u_i(s') - u_i(s) \leq \min_{\pi:\mathcal{S} \to \mathcal{A}} \mathbb{E}\left[\sum_{t=0}^{h_{s \to s'}(M,\pi)-1} r_{\max} - r_t\right].$$

Since $s, s'$ are arbitrary, we can maximize over pairs of states on both sides and get

$$\max_{s'} u_i(s') - \min_s u_i(s) \leq \max_{s,s'} \min_{\pi:\mathcal{S} \to \mathcal{A}} \mathbb{E}\left[\sum_{t=0}^{h_{s \to s'}(M,\pi)-1} r_{\max} - r_t\right] = \kappa(M).$$

It should be noted that even in some cases where the hitting time is infinity—in a non-communicating MDPs for example—$\kappa$ can still be finite and this inequality is still true! In these cases, $r_t = r_{\max}$ except for finitely many terms implying $\rho^*(M,s) = r_{\max}$.  $\square$

## A.2 Proof of Theorem 2

Given an MDP $M$ with finite maximum expected hitting cost $\kappa(M) < \infty$ and an unsaturated optimal average reward $\rho^*(M) < r_{\max}$, the maximum expected hitting cost of any PBRS-parametrized MDP $M^\varphi$ is bounded by a multiplicative factor of two

$$\frac{1}{2}\kappa(M) \leq \kappa(M^\varphi) \leq 2\kappa(M).$$

*Proof.* We denote the expected hitting cost between two states $s, s'$ as

$$c(s, s') := \min_{\pi:\mathcal{S}\to\mathcal{A}} \mathbb{E}\left[\sum_{t=0}^{h_{s\to s'}(M,\pi)-1} r_{\max} - r_t\right].$$

Suppose that the pair of states $(s, s')$ maximizes the expected hitting cost in $M$ which is assumed to be finite

$$\kappa(M) = c(s, s') < \infty.$$

Furthermore, the condition that $\rho^*(M) < r_{\max}$ implies that the hitting times are finite for the minimizing policies. This ensures that the destination state is actually hit in the stochastic process.

Considering the expected hitting cost of the reverse pair, $(s', s)$,

$$\kappa(M) = \max\{c(s, s'), c(s', s)\} \leq c(s, s') + c(s', s) \tag{11}$$

since hitting costs are nonnegative.

With $\varphi$-shaping,

$$c^\varphi(s, s') = \min_{\pi:\mathcal{S}\to\mathcal{A}} \mathbb{E}\left[\sum_{t=0}^{h_{s\to s'}(M,\pi)-1} r_{\max} - r_t^\varphi\right]$$

$$= \min_{\pi:\mathcal{S}\to\mathcal{A}} \mathbb{E}\left[\sum_{t=0}^{h_{s\to s'}(M,\pi)-1} r_{\max} - (r_t - \varphi(s_t) + \varphi(s_{t+1}))\right]$$

By telescoping sums

$$= \min_{\pi:\mathcal{S}\to\mathcal{A}} \mathbb{E}\left[\varphi(s_0) - \varphi(s_{h_{s\to s'}(M,\pi)}) + \sum_{t=0}^{h_{s\to s'}(M,\pi)-1} r_{\max} - r_t\right]$$

By definition of a finite hitting time, $s_{h_{s\to s'}(M,\pi)} = s'$

$$= \varphi(s) - \varphi(s') + \min_{\pi:\mathcal{S}\to\mathcal{A}} \mathbb{E}\left[\sum_{t=0}^{h_{s\to s'}(M,\pi)-1} r_{\max} - r_t\right]$$

$$= \varphi(s) - \varphi(s') + c(s, s') \tag{12}$$

and that the minimizing policy for a state pair will not change. Therefore,

$$\kappa(M^\varphi)$$

By definition of MEHC

$$\geq \max\{c^\varphi(s, s'), c^\varphi(s', s)\}$$

By (12)

$$= \max\{c(s, s') + \varphi(s) - \varphi(s'), c(s', s) + \varphi(s') - \varphi(s)\}$$

The maximum is no smaller than half of the sum

$$\geq \frac{1}{2}[c(s, s') + c(s', s)]$$

By (11)

$$\geq \frac{1}{2}\kappa(M).$$

We obtain the other half of the inequality by observing $M = (M^\varphi)^{-\varphi}$. □