[Reviews · NeurIPS 2019]

Reviewer 1



This paper introduces a new complexity measure for MDPs called maximum expected hitting cost. Unlike the diameter measure is only a function of the transition dynamics, this new measure takes into account the reward dynamics as well. The authors show theoretically that under the same assumptions as previous authors who introduced diameter, this new measure is a tighter upper bound. Furthermore, they show the usefulness of this measure by showing that it can be used to better understand the informativeness of rewards when using potential based reward shaping and they prove theoretically that in a large class of MDPs potential based reward shaping is bounded by a multiplicative factor of 2 on their maximum expected hitting costs. I enjoyed reading this paper. I appreciated the structure that the authors used in this paper which first introduced all the necessary prior work (related to diameter) cosily but thoroughly enough before introducing their contributions. The practical example (Figure 1) with potential based reward shaping was very useful for me to get a better the intuition behind this work and move it from something purely theoretical to something more practical. Originality: Good originality. Given that is only a function diameter of transition, it is natural to ask if there is a better measure that incorporates reward and this was well executed by the authors. Quality: good quality. Theoretical contributions clear and proofs included in the supplementary material. Clarity: excellent clarity. Related work well presented. Paper is self-contained. Contributions are very clear. Significance: good significance. A tighter upper bound to diameter under the same set of assumptions is introduced while the connections to potential based reward shaping are interesting. Furthermore, this paper may be useful in practice to craft better shaping rewards. Minor comments: 1. line 180: The recent rise of deep RL also exposes “bugs” in some of these designed rewards - can the authors cite exactly which kinds of bugs they are referring to here 2. line 181: "Our results show that the informativeness of rewards, an aspect of “the complexity of the learning problem” can be controlled by a well specified potential without inadvertently changing the intended behaviors of the original reward". Read in isolation it might appear that this paper proved that potential based reward shaping leaves the policy unchanged, but this was well-known before. What this paper does is enlighten us on the impact of potential based reward shaping relative to the author’s introduced measure. I think the authors should reword this to make their contributions more clear. 3. line 184: "Rewards should be first defined to faithfully express the intended task, and then any extra knowledge can be incorporated via a shaping potential to reduce the sample complexity of training to obtain the same desired behaviors". Again, I find this statement a bit off-topic from the main paper. It is well-known from potential based reward shaping literature that a shaping reward incorporates domain knowledge and has the potential to learn a policy faster if well specified. I do not see this as being tied to the main contributions of the paper. Rather this paper has enlighting us on the maximum positive / negative impact that potential based reward shaping can have. 4. In general, I felt the final discussion did not tie in with the rest of the paper. It felt like it was talking more about potential based reward shaping and its benefits which is not the main work that is introduced by the authors in this paper. I felt the authors should talk here more about their main contributions and how with their new measure we can start thinking about specifying potential based reward shaping in a more formal way rather than relying on intuition about what is good prior knowledge to include in the potential function. typos: line 66: aglorithm -------- Post rebuttal: Thank you for your comments. I recommend acceptance.

Reviewer 2



This paper proposes the MEHC complexity measure. That MEHC is used to obtain a tighter bound in the context of UCRL2 by a relatively simple modification to the original derivation. To the best of my knowledge, this is an original result. The paper is well-written and provides an interesting insight on the fact that the reward function in itself should ideally be taken into account in complexity measures. The paper might possibly be improved by considering different usage than for improving UCRL2 bounds. The definition of the MEHC depends on r_{max}, wouldn't it be more consistent with the diameter to define it irrespectively of r_{max} by using a scaling factor (i.e., dividing it by r_{max})? ---------------------------------- UPDATE: After reading the author's rebuttal, I see that the authors have provided an interesting discussion based on our suggestions. I have chosen to maintain my score and I believe that the paper is a good paper that should be accepted.

Reviewer 3



The paper proposes a simple measure to replace diameter to improve the regret bound and the sample complexity of UCRL2. As UCRL2 is widely studies algorithm in theory side of RL, the analysis is relevant broadly. The improvement on the bound is rather small on usual reinforcement tasks where the reward is much smaller than rmax for most of the transitions. In particular if the task is sparse reward task where the agent gets rmax on reaching some specified goal and zero otherwise, then MEHC is almost always the same as diameter. It is unclear how much more knowledge we get from using MEHC if we do not have good reward or shaping. Section 2.5 analyzes the effect of shaping on MEHC which is very impressive, but it shows that MEHC can only be halved at best. > Our results show that the informativeness of rewards, an aspect of “the complexity of the learning problem” can be controlled by a well specified potential without inadvertently changing the intended behaviors of the original reward. While the analysis looks very impressive, I do not take it as an evidence to support this claim. In my opinion, a multiplicative factor of two is not impressive enough for many practical applications. As the sample complexity of UCRL2 is linear to MEHC, it only reduce the sample complexity to half. Good heuristic based on expert knowledge can speed up the algorithm much more than that (e.g. intrinsitc reward by Singh et al. 2005). Thus, I would rather understand Theorem 2 as showing the limit of potential-based reward shaping. If we are constrained on using potential-based shaping, then we can only get twice as fast speedup, and in order to get more we have to rely on heuristic which may change the optimal policy. Of course the actual speed up you get seems to be much faster than twice, as shown in the original potential-based shaping paper (e.g. Ng et al. 1999, Figure 1). Overall, while I am very impressed by the bounds shown in Theorem 2, I do not see it as a strong evidence to support potential-based reward shaping. ----------------------------- Thank you very much for clarifying my questions. The author clarified their claim on the impact of reward-shaping which was my only concern on the paper. In the paper at glance it seemed they claimed that "the complexity of the learning problem can be controlled by PBRS" but they clarified in rebuttal that it is the complexity when using UCRL2 algorithms that can be controlled by PBRS. With that qualification it is a valid statement from the theoretical results they provided. I believe the paper has significant theoretical results which should be discussed in NeurIPS. I am very interested in their future work on extending their analysis. With this I will vote for accepting the paper.

[Author Response · NeurIPS 2019]

We thank the reviewers for their thoughtful comments and helpful suggestions, and respond to comments below. We
are pleased that the reviews are enthusiastic. R1: *"Interesting theoretical connections of maximum expected hitting*
*cost to potential based reward shaping . . . This work may be useful in practice to craft informative shaping rewards"*
and *"good significance. A tighter upper bound to diameter under the same set of assumptions is introduced while the*
*connections to potential based reward shaping are interesting. Furthermore, this paper may be useful in practice to*
*craft better shaping rewards."*; R2: *"The paper is well-written and provides an interesting insight on the fact that the*
*reward function in itself should ideally be taken into account in complexity measures."*; and R3: *"the paper improves*
*the regret bound on a very important algorithm by simple analysis"*

**R1:** "bugs" in designed rewards (minor comment 1) – In the particular context, we meant that from the perspective of the
reward designer, the specified reward function might not be consistent with the desired behaviors (see L171). An
example is the commonly cited OpenAI blog post on a faulty reward function in a game called CoastRunners (regrettably
we cannot include any external links in the rebuttal). The quotation marks are to indicate the colloquial use of the term
bug. We appreciate your suggestion and we shall add an explicit reference in any future versions.

**R1:** discussion of contributions (minor comments 2-4) – We agree with your assessment that the discussion section can be
enhanced by emphasizing the *formal* results we established in this work. In contrast, in prior works, PBRS's learning
efficiency was only supported by numerical evidence. We are also excited to mention some prospects to extend this
work in both theory and practice in any future versions.

**R2:** applicability of MEHC beyond UCRL2 – We are also curious about the same question for non-optimism-based algorithms.
We plan to pursue it in future works.

**R2:** $r_{max}$ in the definition of MEHC – This is an interesting suggestion and we will consider it seriously. One benefit we
currently see in keeping $r_{max}$ in the definition is to remind readers (and users) that we assume some knowledge of
$r_{max}$ in the bounded MDP setting (see footnote 2). Note also that a regret bound has a *unit* of "rewards" and even in the
case of the unitless diameter, the resulting regret bound would include $r_{max}$, e.g., $\tilde{O}(r_{max}DS\sqrt{AT})$ from the original
UCRL2 analysis.[1]

**R3:** MEHC and diameter – We agree that in *some* MDPs the gap between these two quantities can be small (or zero).
However, we think this new complexity measure is worth studying as it provides a valuable tool to study the impact of
rewards on learning efficiency as we have shown.

**R3:** limited impact of PBRS on MEHC – We refrain from making a harsh judgment on the merit of PBRS because in this
work, we focus on the average reward setting and UCRL2 whose regret scales with MEHC. It is conceivable that for a
different setting and a different RL algorithm that does not scale with MEHC (or scales poorly with a larger exponent),
such as SARSA with epsilon-greedy exploration as used in [NHR99, footnote 4], PBRS might create a larger impact on
learning efficiency as you suggested. We do agree that the discussion section may be enhanced by pointing out this
future research direction and by adding qualifications lest the claims sound exaggerated.

**R3:** PBRS vs other techniques – As noted in the paper, PBRS is restrictive as the shaped rewards and the original rewards
are Π-equivalent (see L156). It is reasonable to expect a pair of non-Π-equivalent rewards—under some other weaker
notion of equivalence—to have a greater difference in their learning efficiencies (under some algorithm). Furthermore
we want to remark that [SBC05] studies a different RL setting with *salient events* and a comprehensive comparison of
different means to incorporate expert knowledge is beyond the scope of this work.

**R3:** a proof sketch – We tried to make the detailed proofs (included as an appendix in the supplementary material) easy
to follow, but we agree that a proof sketch will further enhance the paper by providing more intuition to readers. In
particular, as you have complimented, Theorem 2 is a very interesting result and our proof, instead of constructing the
best/worst potentials, relies only on the definitions of MEHC and that PBRS does not change rewards on a loop.

## Footnotes

[1] In [JOA10], $r_{max}$ is assumed to be 1 reward-unit making the expression *look* unitless.


[Meta-Review · NeurIPS 2019]

The paper introduces a new complexity measure for MDPs, the expected hitting costs. In contrast to former complexity measures, the hitting costs also depend on the reward of the MDP and can provide a tighter bound for UCRL2. The theory also provides an intersting connection between reward shapeing and the complexity of a MDP. All reviewers appreciated the strong theoretical contribution of the paper which improves our theoretical understanding of the complexity of MDPs. The reviewers also liked that the paper is well written and establishes connections to reward shaping, a method that has also a highly practical value. All reviewers recommend acceptance and I agree with their assessment.